# Unidirectional Magnetic Anisotropy in Molybdenum Dioxide–Hematite Mixed-Oxide Nanostructures

**DOI:** 10.3390/nano12060938

**Published:** 2022-03-12

**Authors:** Felicia Tolea, Monica Sorescu, Lucian Diamandescu, Nicusor Iacob, Mugurel Tolea, Victor Kuncser

**Affiliations:** 1National Institute of Materials Physics, Atomistilor 405A, 077125 Magurele, Romania; felicia.tolea@infim.ro (F.T.); diamand@infim.ro (L.D.); nicusor.iacob@infim.ro (N.I.); tzolea@infim.ro (M.T.); 2Department of Physics, Duquesne University, Pittsburgh, PA 15282, USA; sorescu@duq.edu

**Keywords:** unidirectional anisotropy, exchange bias, coercive field, Morin transition, Mössbauer spectroscopy

## Abstract

MoO_2_-Fe_2_O_3_ nanoparticle systems were successfully synthesized by mechanochemical activation of MoO_2_ and α-Fe_2_O_3_ equimolar mixtures throughout 0–12 h of ball-milling. The role of the long-range ferromagnetism of MoO_2_ on a fraction of more defect hematite nanoparticles supporting a defect antiferromagnetic phase down to the lowest temperatures was investigated in this work. The structure and the size evolution of the nanoparticles were investigated by X-ray diffraction, whereas the magnetic properties were investigated by SQUID magnetometry. The local electronic structure and the specific phase evolution in the analyzed system versus the milling time were investigated by temperature-dependent Mössbauer spectroscopy. The substantially shifted magnetic hysteresis loops were interpreted in terms of the unidirectional anisotropy induced by pinning the long-range ferromagnetic order of the local net magnetic moments in the defect antiferromagnetic phase, as mediated by the diluted magnetic oxide phase of MoO_2_, to those less defect hematite nanoparticles supporting Morin transition. The specific evolutions of the exchange bias and of the coercive field versus temperature in the samples were interpreted in the frame of the specific phase evolution pointed out by Mössbauer spectroscopy. Depending on the milling time, a different fraction of defect hematite nanoparticles is formed. Less nanoparticles supporting the Morin transition are formed for samples exposed to a longer milling time, with a direct influence on the induced unidirectional anisotropy and related effects.

## 1. Introduction

Molybdenum dioxide, MoO_2_, as a nanoparticulate system, has various energy-related applications. Other applications are in the fields of catalysis and electrochemistry [1,2,3,4].

Regarding the magnetic properties, nanoparticles of MoO_2_ could present the diluted magnetism behavior with the diluted ferromagnetic phase persisting above room temperature (RT). It was reported in by Zhang et al. [5] that the functionality of MoO_2_ can be enlarged by mixing it with other functional nano-systems (e.g., MoS_2_ or reduced graphene oxide for increasing the capacity of lithium-ion batteries).

The well-known hematite (α-Fe_2_O_3_) has various applications and can be used as a semiconductor compound [6], magnetic material [7], catalyst [8], and gas sensor [9]. Incorporation of other active ions into the α-Fe_2_O_3_ structure can lead to changes in unit-cell dimensions, particle size, as well as in the magnetic and optical properties. New applications may be developed by combining molybdenum dioxide and hematite in mixed-oxide nanostructures.

The objective of this work is to extend the research on diluted magnetism and to evaluate the possibility to control the RT ferromagnetism of MoO_2_ via the interfacial magnetic interactions with the antiferromagnetic phase of hematite in the equimolar mixtures of MoO_2_ and α-Fe_2_O_3_ nanoparticles. The exchange interactions between a ferromagnet (FM) and an antiferromagnet (AFM) across an interface, responsible for the unidirectional anisotropy, have attracted much attention because they give rise to both an exchange bias field as well as an enhanced coercive field, if compared with the case of a free (uncoupled) FM nanophase [10,11,12,13,14]. The so-called exchange bias systems have been exploited in spin valve devices such as magnetic read heads [15,16], nonvolatile memories [17,18], and various sensors [19,20]. Nanoparticulate systems involving unidirectional anisotropy are also intensively studied due to the possibility to increase the magnetic stabilization (magnetic blocked regime of fine nanoparticles) or the magnetic hyperthermia effects [21,22]. 

It is worth mentioning that this paper continues a series of reports on molybdenum dioxide-hematite nano-systems with different molarities [23,24], this time with the focus being on the observation and the explanation of the unidirectional anisotropy phenomena and related effects in the equimolar nanocomposite systems. 

## 2. Materials and Methods 

Powders (purchased from Alfa Aesar) of MoO_2_ (with an average particle size of about 100 nm) and hematite (with an average particle size of about 50 nm) were subjected to milling in a hardened steel vial with 12 stainless-steel balls of the high-energy SPEX 8000 mixer mill. The mass ratio between the 12 balls and the powder-like equimolar mixture was 5:1. Before introducing the powders in the ball-milling chamber, the powders were manually ground in air to obtain a homogeneous mixture (the 0 h sample). MoO_2_-Fe_2_O_3_ nanoparticle systems were successfully synthesized by mechano-chemical activation through ball-milling of MoO_2_ and α-Fe_2_O_3_ equimolar mixtures for 2, 4, 8, and 12 h of ball-milling time [23,24]

The structural characterization and the size evolution of the nanoparticles were investigated by X-ray diffraction (XRD). XRD patterns have been recorded at room temperature with a PANalytic X’Pert Pro MPO powder diffractometer (with Cu-Kα radiation, λ = 1.54187 Å). The lattice parameters *a*, *b,* and *c* were determined via Rietveld refinement of the XRD patterns, in the hypothesis of the Pseudo-Voigt profile of the lines. Additionally, the average crystallite size was calculated using the Scherrer method.

The magnetic properties were investigated by magnetometry using a Superconducting Quantum Interference Device (SQUID), a Quantum Design magnetometer working under the most sensitive reciprocal space option. 

The local electronic structure and the specific phase evolution in the analyzed system versus the milling time were investigated by Mössbauer spectroscopy. Temperature-dependent Mössbauer spectroscopy between 5 and 295 K has been performed for a full and comprehensive characterization of the atomistic and magnetic configuration of the Fe-based nanophases. The temperature-dependent spectra were collected by inserting the samples in a closed-cycle He flow cryostat. A ^57^Co source in the Rhodium matrix was used on a spectrometer working under the constant acceleration mode. The Mössbauer spectra have been fitted by the NORMOS software package (Brand RA (1987) NORMOS program, Internat Rep Angewandte Phys. Universität Duisburg) and the calibration was performed by using an α-Fe metallic foil at RT. Isomer shift (IS) values, reported relative to α-Fe at room temperature (RT), are given in mm/s, similar to the quadrupole splitting (QS) values. In case of sextet-like patterns, the values of the quadrupole corrections, QS*, are provided. 

## 3. Results and Discussions

The lattice parameters, crystallite size, and phase content of XRD patterns for MoO_2_-α-Fe_2_O_3_, subjected to high-energy ball-milling (HEBM) from 0 to 12 h, are presented in Table 1. The Rietveld structural refinement shows that parameter *c* of hematite presents a slight increase during the milling process due to Mo substituting Fe in the hematite lattice. The crystallite size decreases with the milling time. For example, after a 2 h milling time, the average crystallite size decreased to 39 nm from above 50 nm for α-Fe_2_O_3_ and to 72.5 nm from above 100 nm for MoO_2_. After a milling time of 12 h, the crystallite size decreased down to 17.7 and 50 nm for α-Fe_2_O_3_ and MoO_2_, respectively.

The XRD patterns of the equimolar mixture MoO_2_ + α-Fe_2_O_3_, corresponding to 2 h and respectively 8 h milling times, are presented in Figure 1. Similar to results reported in [23,24], one can notice a decrease of the diffraction peak intensities for both Fe_2_O_3_ and MoO_2_ phases with the increase of the ball-milling time. 

This behavior suggests not only a broadening of the diffraction line, but also a substitution of Fe^3+^ ions with Mo^4+^ ions in the hematite lattice, and vice versa in the molybdenum dioxide lattice. However, the solubility of MoO_2_ into Fe_2_O_3_ is very limited, as resulting from Table 1. The proportion of Fe_2_O_3_ and MoO_2_ is almost insensitive to the milling time (within the error limits) with 64(1) wt.% corresponding to α-Fe_2_O_3_ and 36(1) wt.% to MoO_2_. Note the slight deviation from the equimolar mixture, with an expected proportion of 59(1) wt.% corresponding to α-Fe_2_O_3_ and 41 wt.% to MoO_2_, with the deviation being lower in case of shorter milling times.

The evolution of magnetization with temperature was measured on heating in a 200 Oe applied field (i.e., 0.02 T magnetic induction), after the samples were firstly cooled in the zero field (ZFC curve). The temperature-dependent magnetization in the same field was subsequently measured during the cooling process (FC curve). The ZFC–FC curves of the considered equimolar mixtures differently milled as well as of the initial components are presented in Figure 2. The divergence between the ZFC and FC curves below 350 K, specific to MoO_2_ and highlighted in Figure 2a, infer a superparamagnetic behavior assigned to clustered magnetic defects in MoO_2_ with a large distribution of blocking temperature (even above 350 K). It is important to note that the finite magnetization (although very low) persists up to the highest measuring temperature of 350 K. This supports the long-range magnetic order specific to a diluted magnetic oxide (DMO) and intensively discussed in our works [14,25,26,27]. Such a very weak ferromagnetism is due to the tiny magnetic moments mainly associated to oxygen defects which are interacting at unexpected longer distances by different specific mechanisms (e.g., the bounded magnetic polarons mechanism). 

Figure 2b reveals the specific Morin transition (T_Morin_) of hematite (see the gray squares curve) at about 260 K. The thermal hysteresis associated with first-order thermodynamic Morin transition from a weak ferromagnetic phase at RT to an antiferromagnetic phase below 260 K [28,29] is very well-highlighted, not only for the initial α-Fe_2_O_3_ component of the mixture, but also for the 0 h sample (manually ground). Note that the magnetization in the antiferromagnetic state is still finite, as proof for magnetic defects which are always present in this state. A much lower amplitude of the Morin transition can be observed in the sample exposed to a 2 h milling time from Figure 2c, suggesting a reduced number of true hematite nanoparticles which are involved in the transition. In the samples subjected to longer milling times, the ZFC–FC curves show only weak curvatures without visible Morin transitions. A very interesting aspect is related to the behavior of specific magnetization (per mass unit) in the case of the pure α-Fe_2_O_3_ component and of the equimolar mixture of the 0 h sample (Figure 2b). Expectedly, the specific magnetization of the mixture should be lower in case of the 0 h mixture due to the relatively lower amount of hematite in this sample. If this expectation is fulfilled below the Morin transition, the behavior is opposite above the transition, where the weak ferromagnetic phase of hematite is formed. The only explanation for an enhanced magnetization above the Morin transition is that the weak ferromagnetic phase increases its magnetic contribution along the field direction due to the presence of the DMO phase, which is characterized by its intrinsic long-range ferromagnetic order. If going on to Figure 2c, the specific behavior of the ZFC–FC curves much better resembles the ones of the long-range magnetic order of the DMO system in Figure 2a than the ones of nanosized hematite shown in Figure 2b, except the two orders of magnitude higher values of magnetization specific to the defect antiferromagnetic phase of hematite. Therefore, as will also be subsequently demonstrated by Mössbauer spectroscopy, we may further assume that the main magnetic contribution of samples from 2 to 12 h comes from long-range ferromagnetic interactions of local net magnetic moments in a defect antiferromagnetic (DAFM) phase of hematite, as mediated by the DMO phase. This high-defect phase of hematite does not support any Morin transition and its specific contribution increases with the milling time. 

Hysteresis loops acquired at 5 and 300 K on the considered samples are shown in Figure 3a,b, respectively. It can be observed that in the case of the milled samples, the magnetization at the same high-enough field is increasing with the milling time. At 300 K, the much higher coercive field of pure hematite infers a higher anisotropy of the weak ferromagnetic phase as compared to the antiferromagnetic phase. The presence of a component of high coercivity in the 300 K hysteresis loops of the milled samples where the DAFM phase affords the dominant magnetic contribution supports the fact that a true hematite phase is also present in samples supporting longer mechanical attrition, although the contribution of this component substantially diminishes with the milling time. 

An interesting feature, revealed by the zoomed view of the coercive field domain from Figure 3, is the field asymmetry of the hysteresis loops, both at 5 and at 300 K, suggesting enhanced unidirectional anisotropy effects. This led us to measure the hysteresis curves also after cooling the sample in 5 T. Figure 4a shows the hysteresis loops recorded at 5 K, for the sample subjected to milling for 2 h, after cooling without and respectively in the 5 T applied field. Figure 4b shows the hysteresis curves measured after FC of the 2 h milled sample at different temperatures. As the details of the insets show, the asymmetry of the hysteresis loops is significant. 

For the sample milled for 2 h, the maximum shift of the field after FC is 385(5) Oe (38.5(5) mT) at 300 K and the maximum coercive field is 550(5) Oe (55.0(5) mT) at the same temperature. In the existing literature [10,11,12,13,14], these asymmetries are related to the interfacial exchange interaction between a ferromagnetic phase and an antiferromagnetic phase. As such, the substantially shifted hysteresis loops were interpreted in terms of the unidirectional anisotropy induced by coupling the DAFM phase specific to disordered hematite to the true hematite-like phase of the nanoparticulate mixture. As an intuitive image, the small size and defect hematite nanoparticles which interact strongly with MoO_2_ give rise to the DAFM phase, whereas the rest of the larger and better-formed hematite nanoparticles provide the true hematite-like phase, which can still support the Morin transition. There is an interfacial interaction between these two types of nanoparticles which leads to the unidirectional anisotropy. 

According to [27], the unidirectional anisotropy involves both an increase of the coercive field and the shift of the hysteresis loop by the so-called exchange bias field (usually negative). The temperature dependence of the exchange bias field and the coercive field obtained in both ZFC and FC procedures is presented in Figure 5. 

Within the error limits, the temperature dependences of the exchange bias field as well as of the coercive field obtained after field cooling followed the same trend as after the zero-field cooling. However, after field cooling, these two parameters presented enhanced absolute values, in agreement with the theoretical expectations of the unidirectional pinning mechanisms [30].

Two aspects are clearly pointed out by Figure 5. Firstly, the highest exchange bias and coercive fields, meaning the highest unidirectional anisotropy, were manifested in the 2 h sample, and the average values continuously decreased with the milling time. Secondly, in the mixture milled for 2 h, where the traces for a Morin transition are still evidenced in Figure 2c, the absolute values of the coercive field and exchange bias field increased with the temperature, especially above 200 K, going against the theoretical expectations specific to such dependencies [30,31]. The first aspect can be explained at this point by the assumption that the 2 h sample provided the highest amount of true hematite-like phase, i.e., supporting Morin transition, as illustrated in Figure 2c, and therefore the amount of pinning phase, and hence the pinning effect was maximal in this sample. Moreover, the amount of true hematite like-phase decreased with the milling time, as suggested in Figure 3, and therefore the pinning effects are expected to diminish continuously with the milling time. On the other hand, if looking at the 2 h sample, the anisotropy of the true hematite-like phase was maximal above the Morin transition, and therefore the interfacial pinning effects exerted by this phase are expected to be higher above 200 K. 

Direct support for this explanation is based on the formation of two types of magnetic phases having relative amounts controlled by different milling times, as will be shown in the next section dedicated to Mössbauer spectroscopy investigations.

Temperature-dependent Mössbauer spectra have been collected on two representative samples of milled nanocomposite mixtures: the 2 h sample, where traces for the Morin transition were evidenced in Figure 2c, and the 8 h sample, without any visible trace of such transition in Figure 2c. All the collected spectra, which are shown in Figure 6, have been fitted by three spectral components: two external sextets and one central paramagnetic doublet, the last one being assigned to Fe ions entering the MoO_2_ lattice at the atomic level (counted by the doublet at the lowest temperature) or forming small Fe^3+^ oxide clusters inside this network (counted by the additional relative contribution of the doublet component at 295 K). According to this supposition, about 2% of Fe enter the MoO_2_ lattice at the atomic level after a 2 h milling time and 5% after an 8 h milling time, whereas the rest of the other 3% and 10% of Fe ions, respectively, form small clusters of Fe oxide inside the MoO_2_. However, the exchange bias effects cannot be explained by coupling the diluted FM phase to such antiferromagnetic clusters, while the exchange bias fields decreased with the milling time whereas the amounts of Fe oxide clusters increased with the milling time.

The two sextet components show the following specific behavior: their superposition is very similar below 100 K and is modified at 295 K, therefore inferring a possible magnetic transition between these temperatures. For both samples, at the lowest temperature of 5 K, the best fitting has been obtained via an external sextet with a hyperfine magnetic field, B_hf_, of 54.6 T (and QS* of 0.25 mm/s) and an internal sextet with B_hf_ of 52.8 T (and QS* close to 0.10 mm/s). At 295 K, the two spectral components were characterized by a hyperfine magnetic field of 51.6 T (and QS* close to −0.11 mm/s) and 52.4 T (and QS* close to 0.10 mm/s), with the relative contribution of the 51.6 T sextet being very close to the one of the 54.6 T from the low-temperature spectra. All these spectral characteristics support the assignment of the sextet changing the sign of QS* below 295 K (and with a hyperfine magnetic field of 54.6 T at 5 K and 51.6 at 295 K) to a better-formed hematite phase with Morin transition at above 260 K. The sextet which keeps the value of the quadrupole splitting almost constant versus temperature and presents a very slight decrease of the hyperfine magnetic field with temperature is assigned to an additional defect phase of hematite (as pointed by XRD), which does not support the Morin transition (most probably due to the incorporation of Mo ions into the hematite structure). While the QS* value for this phase is positive and similar to the one of antiferromagnetic phase in pure hematite, a defect antiferromagnetic order involving local net magnetic moments has to be considered for this phase. As a matter of fact, the sextet belonging to well-structured hematite has a relative spectral area of about 57% in the sample milled for 2 h and of only 30% in sample milled for 8 h. The predominance of the pure hematite phase in the sample milled for 2 h also explains the very clearly evidenced Morin transition in the first sample as compared to samples milled for longer times, where the defect hematite phase (without Morin transition) becomes dominant. It is this phase of DAFM hematite which provides the magnetic signal of the samples, of almost similar magnitude at RT and at low temperature. As a direct support for this statement, the magnetization in the low field increased by a factor of about 1.6 in the 8 h sample as compared with the 2 h sample (see Figure 2c), in excellent agreement with the relative spectral contributions of 70% and 43% of the sextets assigned to the DAFM hematite phase in the 8 and 2 h samples, respectively. 

Under the same reasoning, the 2 h sample was characterized by 57% of pure hematite pinning phase and 43% of pinned DAFM phase, in contrast to the 8 h sample, where the relative amount of the pinning phase was much lower. Therefore, the unidirectional anisotropy was the highest in the 2 h sample and diminished with the milling time. Moreover, in the same 2 h sample with the highest relative amount of pure hematite phase, there was the strongest contribution of the higher anisotropy specific to the weak ferromagnetic phase above the Morin transition. Therefore, stronger pinning effects and unidirectional anisotropy were observed above 200 K mainly in this sample. 

## 4. Conclusions

Nanocomposites of α-Fe_2_O_3_ and MoO_2_ interacting nanoparticles were obtained from equimolar mixtures via mechanochemical activation. The average size of hematite nanoparticles continuously decreased with the milling time, i.e., from 50 nm before milling down to 17 nm after 12 h of milling. Additionally, parameter *c* of hematite showed a slight increase during the milling process due to Mo substituting Fe in the hematite lattice. As expected, the structural disorder also increased with the milling time. Magnetic measurements and Mössbauer spectroscopy provide support for the formation of two types of hematite nanoparticles: (i) the less defected ones, denoted as a pure hematite phase which suffers the usual Morin transition, and (ii) the more defected ones, denoted as a DAFM phase which is present over the entire temperature interval and involves long-range ferromagnetic order between local net magnetic moments, as mediated by the diluted magnetic oxide, MoO_2_ nanophase. The substantially shifted hysteresis loops of the nanocomposite mixture were interpreted in terms of the unidirectional anisotropy induced by pinning the long-range ferromagnetic phase specific to the DAFM nanoparticles to the less defect nanoparticles of pure hematite phase. Both the magnetic order and the anisotropy constant of the pinning phase can be tuned by crossing the specific Morin transition. 

Temperature-dependent Mössbauer spectroscopy investigations showed that the relative content of the pure hematite phase and of the DAFM phase, respectively, can be modified by the milling time. In particular, the amount of the pinning phase decreased with the milling time. Consequently, the unidirectional anisotropy and related parameters such as the exchange bias field and the coercive field of the nanocomposite systems decreased with the milling time. 

## Figures and Tables

**Figure 1 nanomaterials-12-00938-f001:**
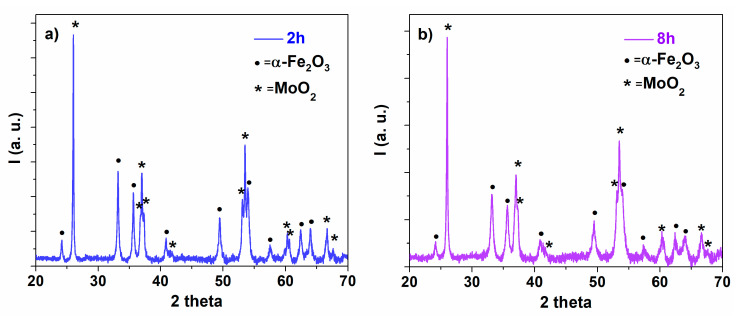
XRD patterns of the equimolar mixture MoO_2_-α-Fe_2_O_3_, corresponding to milling times of (**a**) 2 h and (**b**) 8 h.

**Figure 2 nanomaterials-12-00938-f002:**
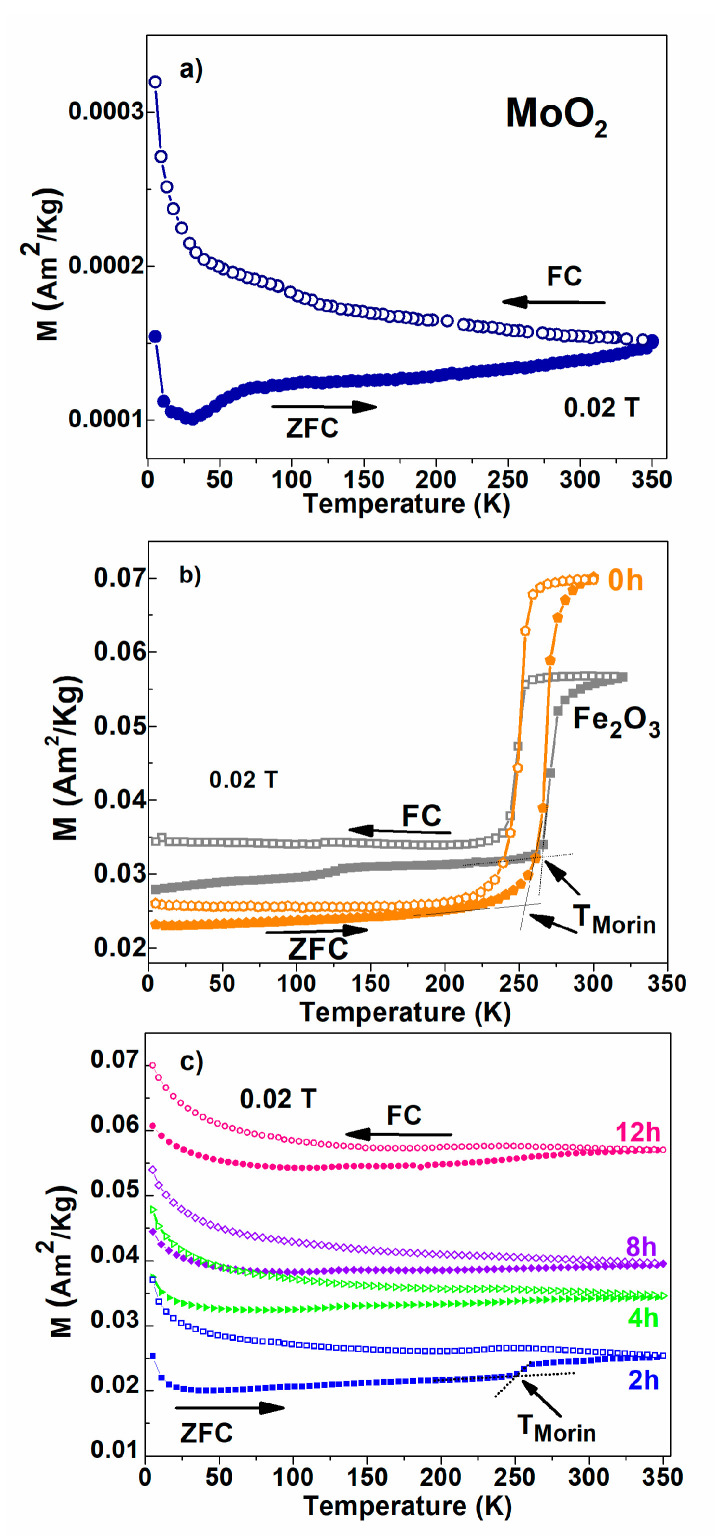
ZFC–FC thermomagnetic measurements, recorded in 200 Oe applied magnetic field (0.02 T magnetic induction), for (**a**) MoO_2_ and (**b**) Fe_2_O_3_ and for the powder mixture manually ground (0 h), (**c**) for samples obtained after 2, 4, 8, and 12 h milling times.

**Figure 3 nanomaterials-12-00938-f003:**
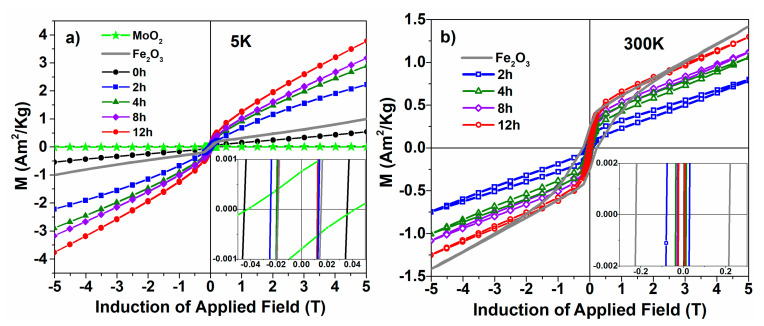
Hysteresis curves recorded (**a**) at 5 and (**b**) at 300 K. Insets: zoomed view of coercivity.

**Figure 4 nanomaterials-12-00938-f004:**
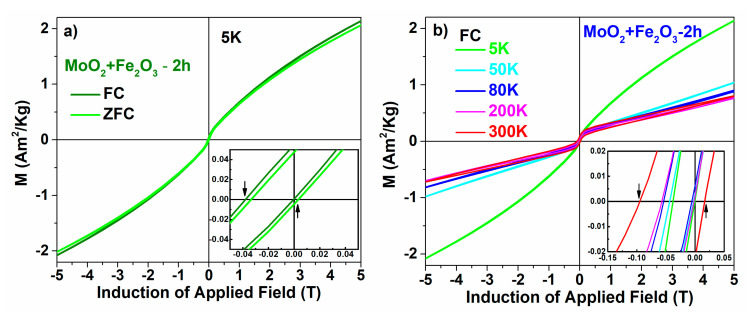
Hysteresis curves for the sample after milling for 2 h (**a**) at 5 K measured on ZFC and FC, and (**b**) measured at different temperatures on FC. Insets: zoomed view of coercivity.

**Figure 5 nanomaterials-12-00938-f005:**
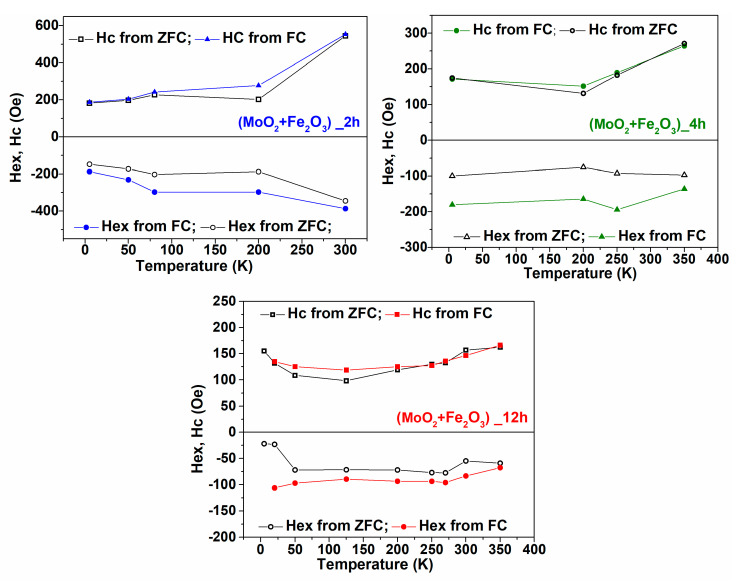
Temperature dependence of the exchange bias field (Hex) and of coercivity (Hc) for samples milled for 2, 4, and 12 h.

**Figure 6 nanomaterials-12-00938-f006:**
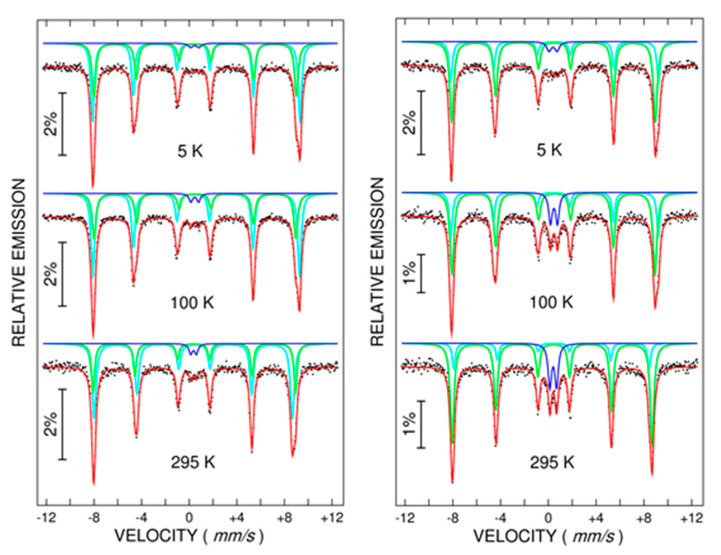
Temperature-dependent ^57^Fe Mössbauer spectra recorded at 5, 100, and 295 K on the equimolar mixture MoO_2_-Fe_2_O_3_ after 2 h (**left side**) and 8 h (**right side**) of milling time.

**Table 1 nanomaterials-12-00938-t001:** The lattice parameters, crystallite size, and phase content from XRD patterns by Rietveld refinement and the Scherrer method for mechano-chemically activated MoO_2_-Fe_2_O_3_ nanocomposites at ball-milling times of: 0, 2, 4, 8, and 12 h, respectively.

Milling Time (h)	Lattice Parameters (Å)			Crystallite Size (nm)	Phase Content (wt.%)
	*a*	*b*	*c*		
0	5.0343	-	13.7430	>50	α-Fe_2_O_3_ (63.9)
	5.6229	4.8541	5.5374	>100	MoO_2_ (36.1)
2	5.0329	-	13.7286	38.9	Mo: α-Fe_2_O_3_ (63.2)
	5.6199	4.8503	5.5372	72.5	Fe: MoO_2_ (36.8)
4	5.0337	-	13.7403	29.2	Mo: α-Fe_2_O_3_ (63.4)
	5.6212	4.8499	5.5408	63.3	Fe: MoO_2_ (36.6)
8	5.0341	-	13.7413	23.1	Mo: α-Fe_2_O_3_ (64.1)
	5.6188	4.8491	5.5447	56.5	Fe: MoO_2_ (35.9)
12	5.0334	-	13.7537	17.7	Mo: α-Fe_2_O_3_ (65.5)
	5.6180	4.8480	5.5457	49.9	Fe: MoO_2_ (34.5)
Statistical Errors	±0.0007	±0.0007	±0.0007	±1.6	±1.3

## Data Availability

Raw data supporting the reported results can be found at NIMP.

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
