# Peer review of "Unidirectional Magnetic Anisotropy in Molybdenum Dioxide–Hematite Mixed-Oxide Nanostructures"

_nanomaterials, 2022, doi:10.3390/nano12060938_

Round 1
Reviewer 1 Report
The authors report on a system of MoO2-Fe2O3 nanoparticle mixture obtained by ball milling. The mixture is characterized by XRD, magnetometry, and Mossbauer techniques. The evolution of the mixture with the milling time is demonstrated also with the temperature dependence of the experimental techniques.
I find this manuscript appropriate for publication in Nanomaterials. I only have few minor comments that I invite the authors to consider:
- In lines 67-68 it is not clear to me what the ratio 5:1 refers to. It may seem that is the ratio between MoO2 and hematite, but this is not clearly the case, since the the mixture is equimolar.
- I notice that there is inconsistency in the units of figures 2,3,4. Magnetization is always indicated in S.I. units, whereas the magnetic field is always indicated in c.g.s units. The same unit system should be chosen.
- In Fig. 2b, the position of the arrows referring to ZFC and FC measurements is misleading, since they are superimposed to orange and grey curves. I would move the arrows away from the curves.
- Something is not clear in Fig. 4. The green curve should be coincident with the blue curve of Fig. 3a. However, the low field enlargements are quite different, since in 4b the exchange bias is much more evident even in the ZFC case. I presume that this is due to the fact that the measurements leading to Fig. 4 were performed after the curves of Fig. 3 were recorded, leading to a sort of “aging” effect. Probably, before recording the curves of Fig. 4, a demagnetization process would have been appropriate. Clearly, the differences between ZFC and FC results in Fig. 4 (but also in Fig. 5!) would have been more remarkable.
- Finally, I found some misprinting or sentences which need to be revised: In line 40, “in” should be removed. In line 95: “decreases” instead of “decrease”. In lines 135-136 there is a sentence in Romanian. In line 164: replace “support” with “supports”. Lines 184-185 should be revised. Line 190: remove “with”. Lines 203-205 check the entire sentence. Line 210: replace “pining” with “pinning”. In line 259 “pinned” is repeated twice. In line 261 replace “diminish” with “diminishes”.
Reviewer 2 Report
In this manuscript the authors use magnetic and Mossbauer studies to explore MoO2 and hematite equimolar -mixtures mixed my milling at various times. The paper provides new information and warrant publication. However, there are several points which must be clarified before.
Page 2 line 68. Powder mass ration 5:1 does not yield an equimolar mixture. It is recommended to use through out the text and in Table 1 the molar value and not the weights. Table 1 shows real mixtures with the same lattice parameters (within the uncertainties)
Page 3 line 105. The decrease in the peak intensities is not visible unless the authors specify real intensity values and not a.u..
Fig. 2a (and the discussion here-after). Bulk MoO2 is not magnetic (see e.g, PRB 81 012402 (2010 and references in there). See also your M(H) of MoO2 at 5 K Fig. 3. Thus the ZFC and FC branches which merge at 350 K (the highest measured temperature) indicates a well high magnetic transition, probably stem from some extra impurities. Note the M values which are two orders of magnitude lower than in Figs 2c. The authors should address this point.
Page 5 line 136. Unclear.
Page 5 line 149. The authors should explain what is the DMO Phase (dilute magnetic oxide)
Fig. 5 (a, b). It is difficult to distinguish between the lines plotted in similar colors (see e.g. Fig. 5c)
The Mossbauer data should be displayed by a Table (with uncertainties). It is difficult to trace the values and changes throughout the text.
The QS deduced for the two sextets are indeed the effective quadrupoles. How were they calculated?
Page 7 line 228 is not clear enough. Are there two Fe-oxide phases in the MoO2 matrix?
Round 2
Reviewer 2 Report
The authors addressed properly all my demands
Author Response
I am very sorry, but I do not see the criticism of the reviewer to the revised version. He mentioned: The authors addressed properly all my demands. So. I do not see what I should have to respond him/her.